# Harnessing metastability for grain size control in multiprincipal element alloys during additive manufacturing

Akane Wakai[1], Jenniffer Bustillos[1], Noah Sargent[2], Jamesa L. Stokes [3], Wei Xiong [2], Timothy M. Smith [3] & Atieh Moridi [1] ✉

Controlling microstructure in fusion-based metal additive manufacturing (AM) remains a significant challenge due to the many parameters that directly impact solidification condition. Multiprincipal element alloys (MPEAs), also known as high entropy alloys, offer a vast compositional space to design for microstructural engineering due to their chemical complexity and exceptional properties. Here, we use the FeMnCoCr system as a model platform for exploring alloy design in MPEAs for AM. By exploiting the decreasing stability of the face-centered cubic phase with increasing Mn content, we achieve notable grain refinement and breakdown of epitaxial columnar grain growth. We employ a multifaceted approach encompassing thermodynamic modeling, operando synchrotron X-ray diffraction, multiscale microstructural characterization, and mechanical testing to gain insight into the solidification physics and its ramifications on the resulting microstructure of FeMnCoCr MPEAs. This work aims toward tailoring desirable grain sizes and morphology through targeted manipulation of phase stability, thereby advancing microstructure control in AM applications.

Multiprincipal element alloys (MPEAs), also known as high entropy alloys, have garnered attention in recent years due to their excellent mechanical and physical properties stemming from their unique chemical complexity. Unlike conventional alloys, MPEAs are composed of multiple elements in high concentration. MPEAs comprising 3d transition metals have been widely explored due to their extraordinary strength and ductility at room[1–3] and cryogenic temperatures[4] owing to their low stacking fault energies and consequent additional deformation mechanisms[5]. In the context of additive manufacturing (AM), MPEAs offer a compelling avenue for the fabrication of high-performance components. Processing MPEAs via AM has led to improvements in properties such as strength[6,7] and corrosion resistance[8] compared to parts fabricated by conventional manufacturing methods such as casting. However, the AM processing of MPEAs presents challenges, just like in conventional alloys, particularly in material systems that solidify along preferred crystal orientations[9].

The high thermal gradients, rapid cooling rates, and partial remelting of previous layers can lead to the growth of columnar grains, resulting in highly textured microstructures. Previous investigations have highlighted the potential of composition as a parameter to alter microstructures in AM[10,11]. However, a deeper understanding of the composition-driven process-structure-property relationships in MPEAs will enable the strategic use of targeted compositions at specified locations to enhance the strength and performance of printed components[12].

Successful examples of alloy design strategies for grain refinement in AM include combining pre-existing alloys such as titanium alloys, nickel-based superalloys, and stainless steels to create new materials[13,14], decorating the powder feedstock with nucleant particles[15], and adding alloying elements to increase constitutional supercooling ahead of the solidification front[16]. In MPEAs, one effective method for changing the microstructure is to introduce a secondary

[1]Department of Mechanical and Aerospace Engineering, Cornell University, Ithaca, NY, USA. [2]Department of Mechanical Engineering and Materials Science, University of Pittsburgh, Pittsburgh, PA, USA. [3]NASA Glenn Research Center, Cleveland, OH, USA. ✉e-mail: moridi@cornell.edu

phase by varying the composition of the alloy, such as adding aluminum, titanium, and niobium to MPEAs with 3d transition metals[17–22]. Secondary phases such as body-centered cubic (bcc), Huesler-like ordered $L2_1$ phase, and Laves phase emerge to pin grain boundaries and break down epitaxial grain growth[23]. However, these additional elements may lead to the formation of brittle intermetallics, which can result in significant cracking during solidification as they cannot accommodate the high residual stresses in AM. In addition, the underlying mechanism by which phase transformation leads to grain refinement remains elusive due to challenges with measuring and interpreting the spatiotemporal evolution of microstructures and defects during and after solidification has finished[24,25]. The current study aims to illuminate a new grain refinement mechanism in MPEAs by combining thermodynamic modeling, operando X-ray diffraction (XRD) studies, and multiscale microstructural evaluation to develop a comprehensive understanding of the intricate microstructural changes occurring during the AM process. We demonstrate an alloy design concept and adaptation for AM using a $Fe_{80-x}Mn_xCo_{10}Cr_{10}$ (at. %, x = 40, 45, and 50) MPEAs. These samples will henceforth be referred to by their Mn content, namely, Mn40, Mn45, and Mn50. In conventionally processed FeMnCoCr, decreasing the Mn content has shown a combination of exceptional strength and ductility due to the lowered stacking fault energy difference between the stable face-centered cubic (fcc) phase and deformation-induced hexagonal close-packed (hcp) phase[1]. Instead, we increase the Mn content to destabilize the primary fcc phase during solidification to enable microstructure control in a direct energy deposition (DED) system where the microstructure generally tends to be more columnar due to its lower scanning speeds compared to powder-bed fusion (PBF)[10]. FeMnCoCr enables us to shed light on the complex relationship of composition, phase stability, process, microstructure, and properties to achieve a crack-free, refined microstructure in AM.

## Results

Phase stability is predominantly influenced by composition, as each element exerts various effects on the overall solidification behavior. In the FeMnCoCr system, the Scheil-Gulliver model[26] predicts the stable phase to be an fcc phase (γ), with a metastable bcc (δ) emerging during the initial stages of solidification (Fig. 1a). Operando XRD studies were conducted at Cornell High Energy Synchrotron Source (CHESS) to reveal the solidification pathways of each composition under the AM condition simulating a DED environment[11,27]. Two detectors, a CdTe Eiger 500k area detector and a far-field GE 41-RT area detector, were placed to capture portions of the diffraction cones at azimuthal angles (η) of $172.4° ≤ η ≤ −172.3°$ (on the left side of the diffracted cone) and $−90.9° ≤ η ≤ 91.8°$ (on the right half of the diffracted cone), respectively (Supplementary Fig. 1). The Eiger detector captured XRD data at 100 Hz to give insight into the fast evolution of diffraction patterns during the AM process, and the GE detector captured data over a larger portion of the cone at a frequency of 4 Hz.

In the time-resolved, azimuthal integration plots from the Eiger data, three peaks associated with fcc are detected at the beginning of the experiment for all compositions (Fig. 1b). As the materials undergo melting, there is a decrease in the 2θ values corresponding to the rise in temperature (thermal expansion). The peaks then shift back to reflect the temperature decrease as the melt pool solidifies and cools down. The original powder peaks in Mn40 and Mn45 remain in the captured data even during melting (between t ≈ 3 s and t ≈ 4.5 s) due to diffraction from the residual powder surrounding the melt pool and deposited bead. These powder peaks during melting are ignored for analysis and interpretation. In the Mn50 sample, an additional peak appears at t = 3.68 s before the fcc peak appears at t = 3.72 s, only to disappear several milliseconds later, as shown in the inset of Fig. 1b-Mn50. This peak corresponds to a metastable bcc phase, confirmed with additional peaks observed on the GE detector annotated in blue

(middle frame during solidification at t = 3.75 s in Supplementary Fig. 2). Unlike all other peaks, which increase in 2θ to reflect cooling, this bcc peak shows a decrease in the 2θ value even though the laser has passed and melting has already occurred (i.e., no more external heating). This peak shift reflects a lattice parameter expansion whose possible causes and implications will be explored later in the Discussion Section. Lastly, there are additional diffraction rings corresponding to the tetragonal $MnO_2$ and $Mn_2O_3$ in all compositions throughout the process (Supplementary Fig. 2).

For a more comprehensive understanding of the solidification behavior, we delve into the raw datasets for information on the evolution of grains in the diffraction condition during the AM process. Azimuth vs. time plots shown in Fig. 1c illustrate the progression of diffraction spots during solidification in the $(220)_γ$ peak as well as the $(211)_δ$ peak for Mn50. Unlike the discrete spots that show up in $(211)_δ$, the fcc spots are more uniformly dispersed. This distinction becomes even more evident when compared against the other two compositions for the same $(220)_γ$ peak. The streaks around 267.5° and 274.9° in Mn40 and those at 276° in Mn45 indicate that there are only a few grains that come into the diffraction condition within the azimuths, while the peaks in Mn50 are lower in intensity (only 4.8% of the maximum intensity of Mn40 and 38.5% of Mn45) and spread out more evenly. The spread indicates a larger number of grains that satisfy the Bragg's condition in Mn50 at various crystallographic orientations, and the weaker signal per peak suggests a smaller average diffracted volume likely stemming from reduced grain sizes.

The microstructural analysis of the single-track beads obtained at CHESS highlight a remarkable trend, as shown in the backscatter electron (BSE) micrographs of Mn40, Mn45, and Mn50 (Fig. 2a–c). Notably, Mn50 exhibits a significant grain size reduction of over 70% compared to the other compositions, with average grain sizes of $20.0 ± 14.8\ μm$, $26.0 ± 21.4\ μm$, and $5.3 ± 3.8\ μm$ for Mn40, Mn45, and Mn50, respectively (Fig. 2d–f), without changing the process parameters. The minimum and maximum grain sizes were $1.6\ μm$ and $112.4\ μm$ for Mn40, $2.7\ μm$ and $115.5\ μm$ for Mn45, and $1.0\ μm$ and $36.0\ μm$ for Mn50, respectively. The directional grain growth towards the build direction in Mn40 and Mn45 is interrupted in Mn50, leaving a more equiaxed microstructure. Comparable grain sizes in their respective powder form (Supplementary Fig. 3) suggests that the cooling rates in DED causes the grain refinement. Phase maps acquired via electron backscatter diffraction (EBSD) show over 99% fcc in all three compositions at these pixel sizes and view fields (Fig. 2g–i). However, BSE imaging unveils the presence of a small fraction of a secondary phase in Mn50 scattered near the edge of the bead. These islands are indexed via EBSD as a tetragonal σ phase, as shown in the inset of Fig. 2c. The σ phase is typically found at grain boundaries and is $2.7 ± 1.9\ μm$ in size. Synchrotron XRD data does not capture any σ phase, which may have occurred because the overall intensities from the σ phase tend to be much weaker than those from the fcc and bcc peaks due to its low-symmetry crystal structure and, therefore, its low multiplicity factor[28], which further increases the difficulty of detecting signal from the σ phase. It is also possible that the volume fraction of the σ phase may be too small to be captured by the detectors. In addition, the expected peaks from the σ phase overlap with many of those in the manganese oxide peaks, making it difficult to distinguish the potential signal from the intermetallic phase. Finally, the σ phase was observed to occur near the bottom of the bead and may not have been within the X-ray window.

Printing the three compositions in multiple layers on a FormAlloy X2 DED system yields a microstructural trend consistent with the single-track beads, as revealed by EBSD shown in Fig. 3a–c. IPF maps depict a columnar grain growth oriented parallel to the build direction in Mn40 and Mn45 shown in Fig. 3a, b, respectively. Such large columnar grains are a direct result of the partial remelting of previous layers often present in as-printed conditions. By contrast, continuous

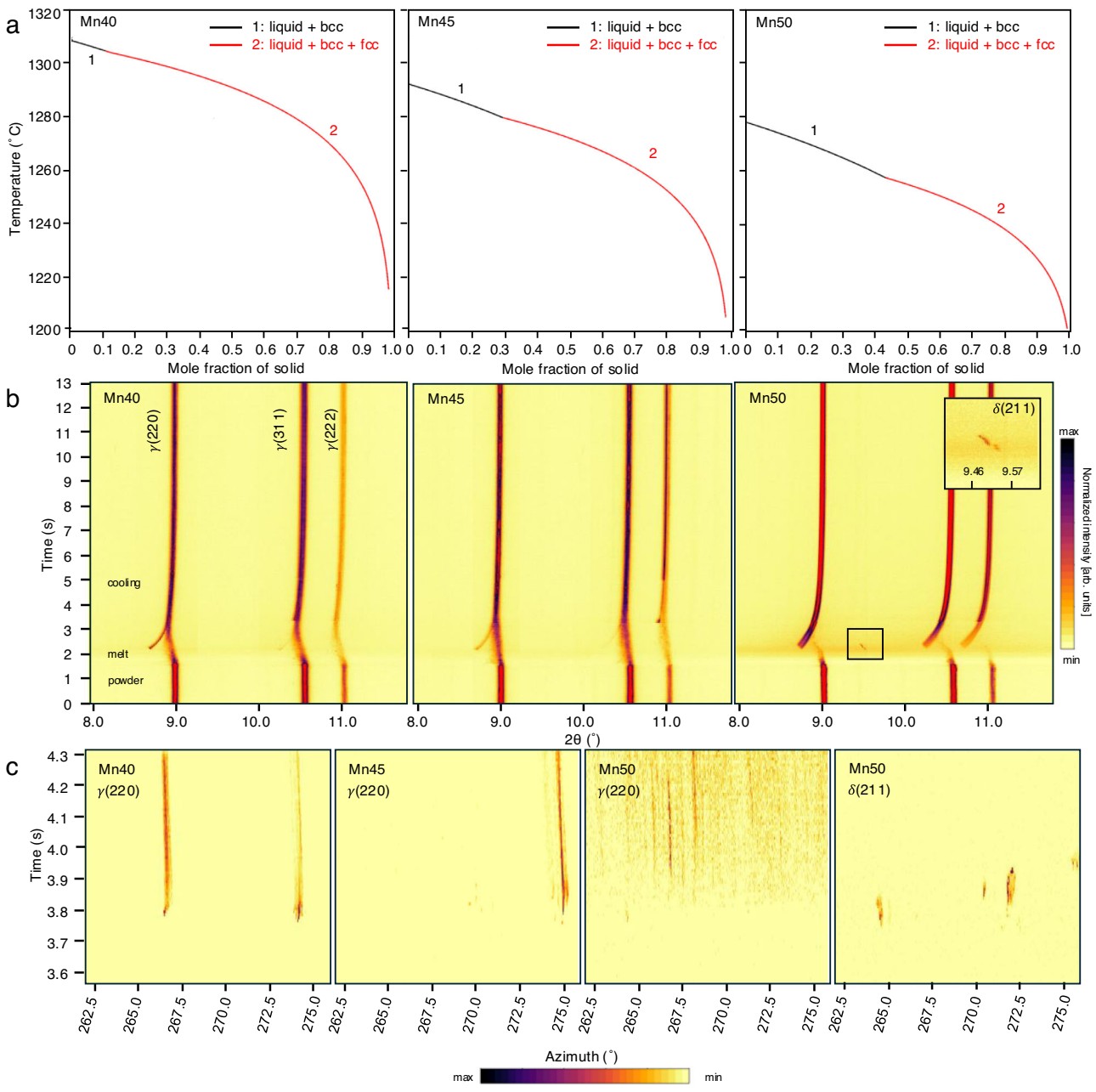

**Fig. 1 | Solidification simulation and operando synchrotron XRD experimental results. a** Solidification pathway simulation of Mn40, Mn45, and Mn50 via the Scheil-Gulliver model using the Thermo-Calc TCHEA6 database[49] predicting a face-centered cubic (fcc) to body-centered cubic (bcc) solidification pathway. **b** 2θ vs. time plots for operando XRD. **c** Azimuth vs. time plots during melting and solidification of the γ(220) and δ(211) peaks for sample Mn50. This plot is representative of other azimuths.

grain growth across layer boundaries is interrupted at the melt pool boundary in Mn50 (Fig. 3c). Moreover, significant grain refinement accompanies the breakdown of epitaxial grain growth, with an average grain size of $41.6 \pm 27.8\,\mu m$ in Mn50 compared to an average of $191.6 \pm 140.8\,\mu m$ and $321.4 \pm 251.7\,\mu m$ for Mn40 and Mn45, respectively. Pole figures shown below Fig. 3c indicate a reduction in texture in Mn50 compared to Mn40 and Mn45, with maximum multiples of a uniform density values decreasing from 3.70 and 3.37 to 2.10 in Mn40, Mn45, and Mn50, respectively.

The presence of σ islands is predominantly observed at the columnar-to-equiaxed transition as indicated by the black box in Fig. 3c. A BSE image of the boxed area shows a combination of the σ phase (bright spots) and the fcc matrix (dark regions). Prominent

chromium enrichment can be found at interfaces between the σ phase and fcc, as indicated by white arrows (Fig. 3f). Most σ islands appear at grain boundaries, as shown in Figs. 2c and 3d. Through scanning transmission electron microscopy (STEM), we observe a significant accumulation of dislocations in the fcc matrix in both Mn40 and Mn50 (Fig. 3e, f, respectively), which is typical of AM parts due to the residual stress from fast cooling and thermal cycling[29].

The changes in chemical composition between the feedstock powders and the as-printed samples were marginal (Supplementary Table 1). At the sub-micron scale, STEM-EDS visualizes the depletion of Fe and Cr and the enrichment of Mn and Co in cellular structures more prominently in Mn40 than in Mn50 (Fig. 3e, f). This phenomenon is commonly observed in as-printed austenitic stainless steels like 304 L

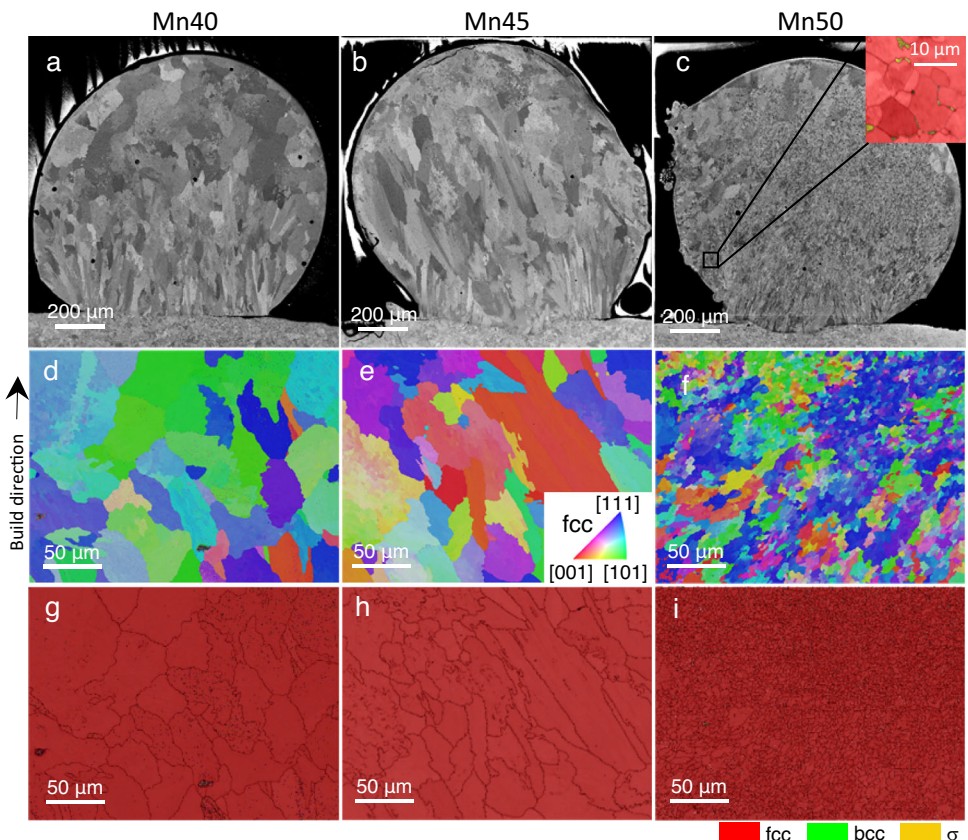

**Fig. 2 | Scanning electron microscopy of single-track FeMnCoCr multiprincipal element alloys. a–c** BSE micrographs of cross sections of single tracks whose operando XRD data were obtained for Mn40, Mn45, and Mn50, respectively. Inset in **c** shows the σ phase (yellow) forming along grain boundaries in Mn50 indexed via EBSD. **d–f** Inverse pole figures (IPFs) of Mn40, Mn45, and Mn50, respectively. **g–i** Phase maps of Mn40, Mn45, and Mn50, respectively. Black lines depict grain boundaries.

and 316L[30] as well as as-printed MPEAs[31]. STEM-EDS also depicts the presence of manganese oxides scattered across both samples (Fig. 3e, f), which has also been observed in other as-printed MPEAs, such as the CoCrFeMnNi[32]. The oxides in Mn40 are significantly larger than those in Mn50, which corresponds with the higher intensities of signal from the oxides in the operando XRD data in Mn40 shown in Supplementary Fig. 2. The initial oxygen composition of the feedstock (likely existing as an oxide film on the surface of the feedstock powder or oxide inclusions within the feedstock) may provide the oxygen to form the fine inclusions. Content of minor elements (O and N) were measured via inductively coupled plasma spectroscopy and are presented in Supplementary 2, which shows 0.0574 wt%, 0.0533 wt%, and 0.0364 wt% oxygen in Mn40, Mn45, and Mn50 powders, respectively. In addition, any oxygen in the atmosphere may be picked up during the print processes. These oxide inclusions nucleate and grow when the solubility of oxygen dissolved in the molten pool decreases[30].

Tensile testing was conducted to assess the mechanical performance of the as-printed MPEAs in a direction perpendicular to the build direction (Fig. 4). Notably, the yield strengths exhibited an upward trend, rising from 372.7 ± 10.8 MPa, 378.2 ± 11.6 MPa, to 411.9 ± 18.3 MPa in Mn40, Mn45, and Mn50, respectively. This enhancement in strength, particularly in Mn50, can be attributed to the grain refinement according to the Hall-Petch relationship[33]. However, there is a slight decrease in ductility with increasing Mn content (28.9 ± 4.1%, 27.9 ± 3.1%, and 26.8 ± 1.0%, respectively). While the Petch-Stroh criterion suggests that the strengthening should be accompanied by increased ductility, the findings show otherwise. There is currently no explanation of the observed trend. SEM analysis of the fracture surfaces reveals dimpled structures across all three

compositions, indicating a ductile fracture (Fig. 4b–d). Though the fracture surface of Mn50 predominantly exhibits dimples, a very small portion shows a serrated fracture characteristic, which suggests brittle failure in likely the sigma phase (Fig. 4d). While the sigma phase appear sporadically and does not seem to cause catastrophic embrittlement, observations suggest a still slight compromise in the overall ductility of the printed structure.

## Discussion

A compelling correlation emerges between the destabilization of the fcc phase and grain refinement. In Mn40 and Mn45, where the fcc phase stability is relatively high, the liquid directly solidifies into fcc along with manganese oxides (Fig. 5a–d). Conversely, Mn50 undergoes a more complex solidification pathway, as illustrated in Fig. 5e–h. The liquid phase first solidifies into bcc (Fig. 5e) as observed in synchrotron XRD data, where the grains may be large and uniquely oriented based on the spotty diffraction spots in azimuth vs. time plot (Fig. 1c). The oxides may also form before or during this time[30]. Subsequently, transformation of fcc grains occurs from the bcc phase, as illustrated in Fig. 5f. The azimuth vs. time plot (Fig. 1c) shows a sudden appearance of uniform diffraction signal from the fcc grains shortly after the onset of signal from the bcc phase. A similar mechanism is observed to break down grains in Fe-C via time-resolved X-ray imaging, where a massive δ-to-γ transformation results in the fragmentation of grains[34]. In addition, we hypothesize that the phase transformation may cause recalescence – the release of latent heat due to phase transformation – which may result in grain remelting to further enhance grain refinement. As mentioned previously, a peak shift towards a lower 2θ value is observed in the bcc peak, which signifies an increase in the lattice

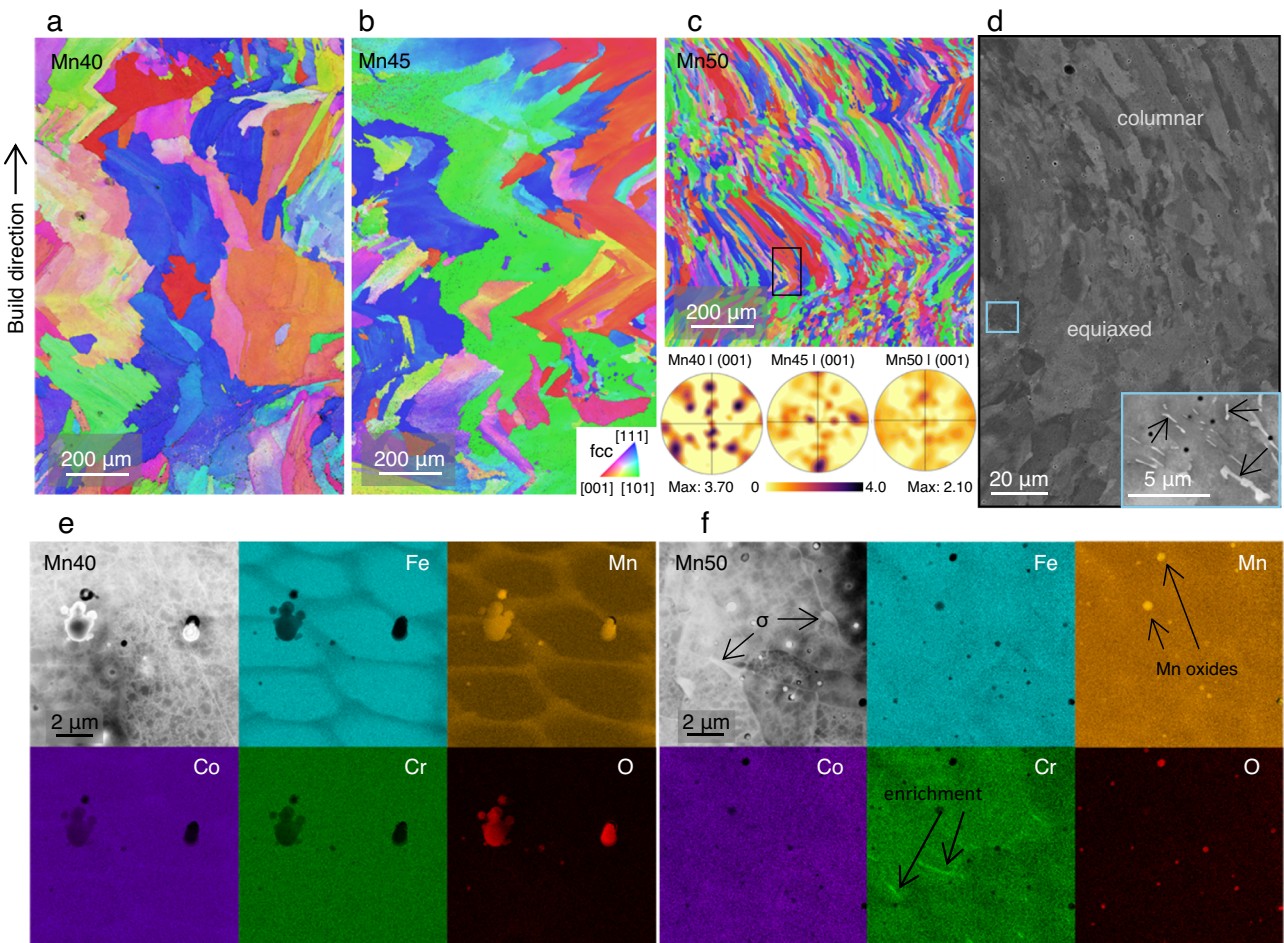

**Fig. 3 | Electron microscopy of as-printed FeMnCoCr MPEAs. a–c** IPFs of Mn40, Mn45, and Mn50, respectively. Pole figures of IPFs in the (001) direction are shown below. **d** BSE image of columnar-to-equiaxed transition showing σ-phase precipitation. **e, f** STEM and EDS images of Mn40 and Mn50, respectively. The electron beam is parallel to the [110] zone axis of the matrix in both STEM images presented. High density of dislocations is seen in both materials. Secondary phase appears brighter than the fcc matrix in Mn50. Cr segregation (white arrows) correspond to interfaces between fcc and sigma phase.

parameter. Multiple possible factors can contribute to this peak shift such as mechanical strain (due to a macrostress)[35], compositional change (e.g. doping[36]), and a rise in temperature[37], and it is difficult to decouple each one from the other. Although we are unable to completely rule out mechanical and chemical contributions to this observation, thermal effects are currently hypothesized to be a major contributor to the peak shift. In a first approximation, we assume that thermal effects are most dominant immediately following solidification when the cooling rates are the highest at ~1000 K/s (i.e. assume no contributions from mechanical and chemical effects), the maximum possible temperature rise detected by the synchrotron XRD would be 107.0 ± 16.8 °C (calculation method is detailed in the Methods Section). Previous studies have shown that δ-dendrite fragmentation can occur when dendrite arms remelt after recalescence during solidification from an undercooled melt[38,39], and remelting has led to grain refinement for an undercooling between 60 °C and 100 °C in Ni-Cu systems[40]. Given that the estimated temperature rise is of a similar magnitude to the recalescence-grain refinement phenomenon, it is plausible that the increasing temperature in the bcc phase could have also remelted the dendrite arms to cause grain refinement as illustrated in Fig. 5f, g. Further rigorous and systematic investigations are needed to isolate effects from chemical, mechanical, and thermal effects on the resulting XRD data to validate this hypothesis.

The solidification pathway of the σ phase cannot be inferred from the operando XRD data because its signal could not be detected.

However, we hypothesize that the intermetallic phase nucleates and grows from the intermediate bcc phase as often seen in duplex stainless steels. The bcc crystal structure is known to transform to the σ phase more quickly than fcc into σ due to a number of factors such as bcc allowing for faster diffusion (almost 100 times greater[41]) in its less densely packed crystal structure[42], the crystallography of the σ phase and fcc are incoherent[43], and bcc and the σ phase are both stabilized by Cr[44]. Although the σ phase has also been shown to precipitate from the fcc phase in some 3d transition metal MPEAs, it occurs only after hundreds or thousands of hours of heat treatment[45] or in samples subjected to severe plastic deformation and recrystallization through annealing[46]. In one reported case, a small amount of the σ phase is found in an as-printed PBF CoCrFeMnNi, where its occurrence is attributed to the small grain size and high density of dislocations that enhance overall diffusion coupled with the repeated thermal cycling in the AM process[6]. In this study, since the σ phase is generally stable between 600 and 1000 °C[43,47], it is inferred that the bcc-to-σ transformation occurs in solid-state (Fig. 5h). The mechanism and kinetics of the intermetallic phase formation in this MPEA systems are subject to further investigation.

Lastly, we note the limitations of both experimentation and simulations conducted for this study. Firstly, the operando XRD data suggests that the formation of the bcc phase occurs only in one composition, though the Scheil-Gulliver model predicts that all of them would undergo a bcc-to-fcc solidification pathway. It is worth

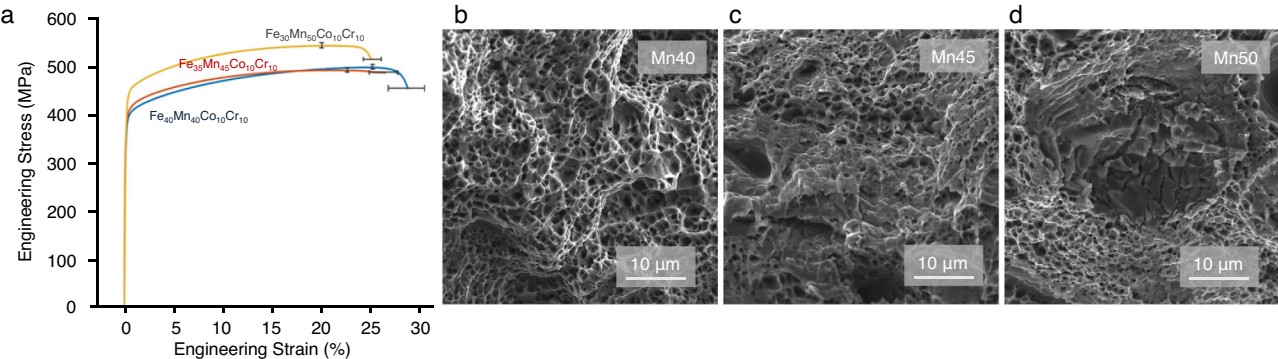

**Fig. 4 | Mechanical behavior and response of MPEAs. a** Engineering stress vs strain curve of as-printed FeMnCoCr MPEAs. Error bars represent one standard deviation from three samples per composition. **b–d** SEM images of fracture surfaces of Mn40, Mn45, and Mn50, respectively. Source Data is provided with this paper.

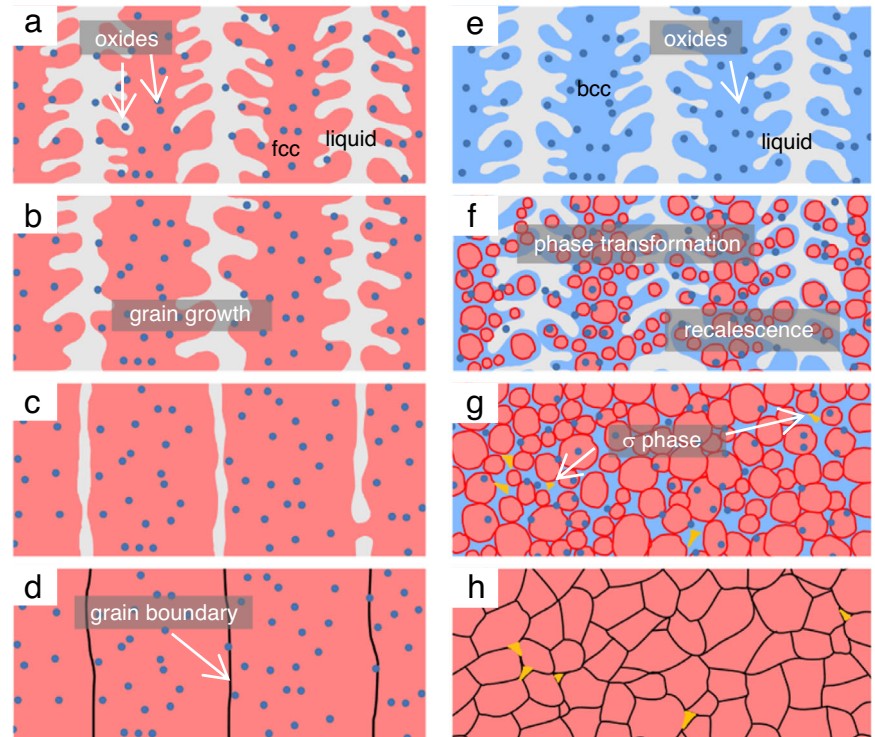

**Fig. 5 | Proposed solidification mechanism. a–d** Solidification and growth mechanism of Mn40 and Mn45. The molten metal solidifies into the stable face-centered cubic (fcc) phase along with oxide particles. With nothing to hinder its growth, these grains grow epitaxially across layer boundaries to promote large,

columnar grains. **e–h** Solidification and transformation mechanism of Mn50. The intermediate body-centered cubic (bcc) phase may act as a grain refiner through transformation to inhibit grain growth with each deposited layer. Recalescence may cause grain remelting to further promote grain refinement.

noting that the current analysis is conducted on a single projection XRD, i.e., no sample rotation, which limits the number of grains aligned in the diffraction condition. In addition, it is possible that bcc may form in Mn40 and/or Mn45 that goes undetected by the detectors if (1) the timescale of bcc formation and transformation in Mn40/Mn45 is faster than the temporal resolution of the current study and (2) the signal from bcc phase is too low to be detected. Setup modifications such as integrating multiple detectors or improvements to beamline and detector characteristics by increasing beam flux is necessary for further probing the fast evolution of microstructures and defects in AM. On the flip side, the Scheil-Gulliver model does not consider solidification rates, which can greatly vary in AM systems (between $10^3$ K/s-$10^5$ K/s in DED and $10^4$ K/s-$10^6$ K/s in PBF systems[10]). Since solidification rates play a crucial role in determining the solidification pathway, relying solely on the Scheil-Gulliver solidification simulation may not always provide an accurate guideline for selecting metastable MPEAs

suitable for all AM techniques. However, the trend found through Scheil-Gulliver – where the bcc formation increases from Mn40, Mn45, to Mn50 – remains valid. In light of these findings, the Scheil-Gulliver model can serve as an initial screening tool for compositions with a transient phase formation during solidification, but additional experimental investigations are necessary to accurately predict and validate the solidification behavior to design MPEAs for AM.

Phase stability engineering is demonstrated as a viable methodology for designing alloys specifically for AM by harnessing its unique processing conditions. The combined operando synchrotron XRD, thermodynamic modeling, microstructural analysis, and mechanical testing elucidates the composition, process, microstructure, and properties relationship. It is worth mentioning that the grain refinement has not been observed between MPEAs of similar composition, namely $Fe_{30}Mn_{50}Co_{10}Cr_{10}$ and $Fe_{50}Mn_{30}Co_{10}Cr_{10}$, fabricated via PBF[48]. This finding suggests that the grain refinement mechanism is highly

dependent on the solidification rate, which is higher in PBF than in DED. The effect of solidification rates on metastable phase formation is subject to further study. These MPEAs exploit the intrinsic rapid solidification through phase transformations of metastable phases that break up continuous grain growth across layer boundaries. These findings set the stage for theory-guided exploration in the extensive compositional space of MPEAs for AM.

## Methods

### Solidification simulation

Scheil-Gulliver simulations[26] were performed using Thermo-Calc[49] software version 2023a with the TCHEA6 thermodynamic database to predict the solidification behavior of each alloy composition. The Scheil model leverages thermodynamic data to predict the phase evolution and microsegregation that occur during solidification. The Scheil-Gulliver model assumes homogeneity in the liquid composition, no diffusion in the solid phase(s), and equilibrium at the solid-liquid interface. Considering the Marangoni convection in AM that effectively homogenizes the melt pool[9], the assumptions made in the model are justified. Previous studies in the literature have shown agreement between simulations and experimental results[50].

### Operando synchrotron X-ray diffraction

Synchrotron X-ray diffraction studies were conducted at Cornell High Energy Synchrotron Source (CHESS). A custom printer was integrated at the Forming and Shaping Technology ID3A (FAST) beamline[27]. A high-energy, monochromatic X-ray beam with a wavelength of 0.2022 Å and energy of 61.322 keV was used in transmission mode with a square cross-section of 0.750 mm×0.750 mm. A CdTe Eiger 500k area detector with 512 ×1024 pixels captured diffraction patterns at a frame rate of 100 Hz and covered azimuthal angles ($\eta$) between −172.3° and 172.4° and diffraction angles ($2\theta$) between 7.5° and 11.2°. The detector-to-sample distance was 899 mm, which was calibrated using a $CeO_2$ reference powder. Additionally, a GE 41-RT area detector with 2048 ×2048 pixels was used to capture azimuthal angles between −90.9° and 91.8° and diffraction angles between 0.6° and 15.4° at a frame rate of 4 Hz.

Powder from each composition was rastered with a 500 W continuous wave multi-mode laser from IPG Photonics at a laser power of 200 W, scanning velocity of 4.5 mm/s, and layer height of 2 mm. These parameters were chosen to achieve a stable bead with the constraint of the scanning speed, which was the maximum velocity attainable on the Huber stage available at CHESS. Azimuthal integration and azimuth vs time plots of the Debye-Scherrer diffraction patterns from the Eiger detector was performed using GSAS-II[51]. For azimuth vs time plots, $2\theta$ values from 8.7° to 9.1° and 7.6° to 7.9° were binned and integrated for the $\gamma(220)$ peaks and $\delta(211)$ peak in Mn50, respectively.

The calculation of temperature rise in the bcc phase was done by first fitting each bcc diffraction spot to extract the $2\theta$ values from the Eiger detector on GSAS-II using the Pseudo-Voigt model. The corresponding d-spacings were calculated using the Bragg's Law, $n\lambda = 2d\sin(\theta)$. Together with the coefficient of thermal expansion (CTE), the change in temperature ($\Delta T$) was acquired using the following equation[52]:

$$\Delta T = \frac{d_f - d_0}{d_0 * CTE} \quad (1)$$

Where $d_0$ and $d_f$ are the initial and final d-spacings, respectively. The reported uncertainty is taken as the standard deviation of the changes in temperature for all diffraction spots.

### Direct energy deposition

FeMnCoCr spherical powders were gas-atomized (Arcast, Oxford, ME) and sieved to achieve particle diameters ranging between 15 and

45 μm. Printing was performed on a FormAlloy X2 DED system equipped with an IPG Photonics Nd:YAG continuous wave fiber laser with 500 W maximum power at a spot size of 1.2 mm. The build chamber was purged with argon to reduce the oxygen level to below 100 ppm. Blocks of 26 mm × 10 mm × 6 mm (WxLxH) were printed on a 304 L stainless steel substrate with a rectilinear infill scan strategy with hatch spacing of 0.6 mm and an angle offset of +67° and −67° in alternating layers with a 0.2 mm layer height. All compositions were printed under the same nominal processing conditions (laser power of 250 W, scanning speed of 800 mm/min, powder feed rate of 0.5 rpm).

### Scanning electron microscopy (SEM)

Single beads from Cornell High Energy Synchrotron Source (CHESS, Ithaca, NY) and as-printed samples were cross sectioned using a high-speed diamond saw parallel to the build direction and polished down to 0.08 μm colloidal silica for compositional and microstructural evaluation. Imaging and energy dispersive spectroscopy (EDS) were conducted on a Tescan Mira3 field-emission scanning electron microscope (FE-SEM) equipped with a backscattering detector. Electron backscatter diffraction (EBSD) was acquired on a QUANTAX EBSD for grain morphology and phase makeup evaluation (Bruker, Billerica MA). The field of view of 300 μm and pixel size of 0.5 μm were used for all single bead EBSD measurements. The field of view of 1000 μm and pixel size of 0.6 μm were used for multilayer Mn50, and a wider view field of 1500 μm and a pixel size of 1.2 μm were used for multilayer Mn40 and Mn45 to capture the larger grains. The measurements were processed using ATEX (Metz, France).

### Transmission electron microscopy (TEM)

Scanning transmission electron microscope (STEM) disc samples (diameter 3 mm) were extracted via wire-electrical discharge machining from Mn40 and Mn50. STEM discs were thinned manually to 100 μm using SiC polishing paper. The STEM discs were then electropolished with a 90% methanol and 10% perchloric acid at −40 °C and 12 V using a Struers twin-jet polisher. STEM-EDS was conducted on a FEI Talos at 200 kV using a Super-X energy dispersive X-ray spectroscopy detector and processed using Thermo Fisher Scientific Velox software.

### Coefficient of thermal expansion

The coefficient of thermal expansion (CTE) was determined via in situ diffraction[53] in an Empyrean diffractometer (Malvern PANalytical) using cobalt radiation (Co K$\alpha$ = 1.789 Å) and a hot stage (Anton Paar HTK 2000N). National Institutes of Standards Technology (NIST) standard reference material (SRM) aluminum oxide ($Al_2O_3$) powder was used as a calibration standard dispersed onto a sacrificial piece of tantalum foil on the tantalum heating strip to determine the temperature set points. The measured values for $Al_2O_3$ were compared to the recommended values for the thermal expansion ($\Delta L/L_0$) as a function of temperature $T$[54], as

$$(a_0)\frac{\Delta L}{L_0} = -0.176 + 5.431 \times 10^{-4}T + 2.150 \times 10^{-7}T^2 - 2.810 \times 10^{-11}T^3 \quad (2)$$

$$(c_0)\frac{\Delta L}{L_0} = -0.192 + 5.927 \times 10^{-4}T + 2.142 \times 10^{-7}T^2 - 2.207 \times 10^{-11}T^3 \quad (3)$$

where temperature is measured in Kelvin. Equations (2) and (3) are then inverted, and the measured values of the average change in lattice parameters are used in the inverted equation to find the actual temperature of the heating strip. Once temperature set points were determined, the sample powders were dispersed onto tantalum foil on the heating strip. Scans were taken over a $2\theta$ range of 40° to 130° at

temperatures of 25 °C, 200 °C, 400 °C, and 600 °C. All scans were analyzed in TOPAS[55] using the LeBail method[56] to estimate the lattice parameters of the materials at each temperature. The average bulk CTE is approximated as 1/3 of the unit cell volume expansion coefficient (i.e., $\Delta V$ vs. $\Delta T$ divided by 3). The value of the coefficient of thermal expansion in Mn50 was determined to be $18.38 \times 10^{-6}$ K$^{-1}$.

## Mechanical characterization

The printed samples were machined perpendicular to the build direction via wire-EDM into micro-tensile specimens with a gauge length of 8 mm, width of 2 mm, and thickness of 0.7 mm. The specimens were ground down to 0.6 mm thickness to remove the oxide layer on both sides. The evaluation of the mechanical properties of the samples was conducted under tensile loads using a Deben MT 2000 micro-tensile stage equipped with a 2 kN load cell (Deben UK Ltd, Suffolk, UK). Tensile experiments were performed in displacement control mode at an average strain rate of $2.0 \times 10^{-3}$min$^{-1}$. Non-contact, real-time evolution of strains was captured by a digital image correlation software (GOM, Braunschweig, Germany) from the recorded tensile displacements. Three tensile samples for each composition were tested to confirm reproducibility, and the average values and standard deviation are reported. The yield strength was determined with the 0.2% offset plastic strain method.

## Data availability

The data that support the findings of this study are available from the corresponding author upon request. Source data are provided with this paper.

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

## Acknowledgements

A.W. thanks Teri Juarez and Bryan McEnerney for discussions and insights at Jet Propulsion Laboratory and beyond. A.W. and A.M. acknowledge the contributions of Katherine Shanks and Amlan Das during experiments at CHESS and on data analysis. A.W. and J.B. thank Chenxi Tian, Jinyeon Kim, Dasol Yoon and Claire Matthews for assistance with sample preparation. A.W. and A.M. thank Kaushalendra Singh and Michael Kulis for their help acquiring additional data on powders. This work is primarily supported by the U.S. Department of Energy, Office of Science, Basic Energy Science Early Career Award #DE-SC0022860, A.M. (Synchrotron data analysis and microscopy) and by the National Science Foundation CAREER Award #CMMI-2046523, A.M. (conducting operando experiments). A.W. acknowledges a graduate fellowship through the NASA Space Technology Graduate Research Opportunities [Grant #80NSSC20K1199, AW]. This work made use of the Cornell Center of Materials Research Shared Facilities, which are supported through the NSF MRSEC program [DMR-1719875]. This work is based upon research conducted at the Center for High Energy X-ray Sciences (CHEXS), which is supported by the National Science Foundation [DMR-1829070]. W.X. acknowledges funding from the National Science Foundation CAREER Award [CMMI 2047218, W.X.]. N.S. acknowledges a graduate fellowship through the NASA Space Technology Graduate Research Opportunities [Grant # 80NSSC19K1142, N.S.]

## Author contributions

A.W. wrote the manuscript. A.W. and A.M. designed experiments. A.W. and J.B. performed experiments at CHESS. N.S. and W.X. performed thermodynamic simulations. A.W. and T.S. performed TEM analysis. A.W. performed microstructural characterization, mechanical testing, and synchrotron X-ray diffraction analysis. J.S. performed in-situ X-ray diffraction. A.W. and A.M. obtained funding. A.M. supervised the work.

## Competing interests

The authors declare no competing interests.
