## [Transparent Peer Review file · Nature Communications]

Harnessing metastability for grain size control in multiprincipal element alloys during additive manufacturing

Corresponding Author: Professor Atieh Moridi

Version 0:

Reviewer comments:

Reviewer #1

(Remarks to the Author)

The present manuscript reports on alloying-related effects in a multi-principal element system and its interactions with additive manufacturing conditions. The reported results are new and interesting and the conclusions drawn are presented in a compelling manner. There remain some minor thoughts outlined below that the authors should consider prior to acceptance of the article.

1. It is unusual to use references in the abstract unless the work specifically refers to any hypothesis of these references. Please consider removal of the references from the abstract or, if required, use the complete reference and not just numbers.
2. Mechanical properties: The authors observe strengthening with a reduction of grain size and relate it to Hall-Petch hardening. According to the Petch-Stroh criterion this should be accompanied by an increase in ductility. I do not see any critical assessment of this apparent contradictory result. Simply stating that it is matter of ongoing investigations, does not seem sufficient for the reviewer.

Reviewer #2

(Remarks to the Author)

This work has assessed the microstructural and mechanical effect by consistently swapping Mn for Fe in a typical HEA system (FeMnCoCr) for DED. The striking finding was a significant refinement of the grain size and also an increase in strength with a trade-off in ductility when Mn was increased to 50%. The manuscript requires some clarification and strengthening before publication.

1. The bcc to fcc transformation is interesting, but the only physical evidence presented was a very brief capture of the bcc (211) reflection spot in the synchrotron data. The authors should show the volume fraction vs temperature plot for bcc and fcc. Hereby to strengthen the proposed mechanism. In addition, I was expecting to see (310) reflection as well. Is there any reason that it was not shown?
2. One further trend that I was confused was the reflection intensities for 311 and 222 in Mn40 and 45 vs 50 (Fig 1 b). They are not following the same trend in 2th as 220 as it cools down from the melt, why? Rather, they show a linear transition from powder to cooling. Meanwhile, Mn50 was consistent with all the reflections shown.
3. Since oxides were frequently observed, the authors should disclose the oxygen level in the starting powder. Other relevant trace elements for C, S, N will be helpful, too. How was the chemical composition measured in the extended data? It needs to be disclosed.
4. I had a quick search in the literature, which I found other groups publishing on the same composition but did not see the same grain refinement trend by increasing Mn (with the same processing conditions, etc). What would be the reason? It deserves a good discussion. An example is here: <https://doi.org/10.3390/mi15010123>
5. The authors should also provide sufficient level of characterisation for powder microstructure, such as the starting grain size, to make the story more convincing.
6. I don't think as-cast structure in Fig 4 should be there, as the comparison is not very fair between casting vs DED. As the literature in AM has shown higher yield strength of the same composition, such as the one above (LPBF though).

Version 1:

Reviewer comments:

Reviewer #1

(Remarks to the Author)

The authors have provided additional information on the fracture behavior but do not provide a compelling theory on the observed reduction in ductility. While I do not see, how this can be argued simply ignoring the absence of an attempt remains insufficient. If the authors do not want to claim a responsible effect, they should state that there is presently no explanation for the observed behavior.

Reviewer #2

(Remarks to the Author)

The authors have addressed my questions appropriately and revised the manuscript accordingly. The overall methodology was elegant and I am convinced the conclusions are sound. I would encourage the authors to disclose when the delta phase disappeared from the diffraction spots on the GE detector. Currently it was only termed as 'quickly disappear' where quantifying the temporal evolution, despite a much lower acquisition (and uncertainty), would help estimate the transformation temperature and interpret the nature of the transformation -- whether it was fully solid state or involving liquid as proposed.

By way of readability, some optional suggestions here. Consider moving Fig 1a to extended data -- this was not so critical in my opinion and it has caused some confusion as I read. You can also consider moving extended data Mn50 evolution to Fig 1 as that is more convincing with rings of delta reflections.

RESPONSE TO REVIEWERS' COMMENTS

Akane Wakai, Jenniffer Bustillos, Noah Sargent, Jamesa Stokes, Wei Xiong, Tim Smith, and Atieh Moridi

Dear Reviewers,

We greatly appreciate both reviewer's thoughtful comments. We are thrilled the reviewers recognize the novelty of the work and hope that the modifications made to the manuscript clarify the confusions and questions. Please find our response below and changes to the manuscript highlighted in yellow.

Very Respectfully,

Atieh Moridi

Reviewer #1 (Remarks to the Author):

The present manuscript reports on alloying-related effects in a multi-principal element system and its interactions with additive manufacturing conditions. The reported results are new and interesting and the conclusions drawn are presented in a compelling manner. There remain some minor thoughts outlined below that the authors should consider prior to acceptance of the article.

1. It is unusual to use references in the abstract unless the work specifically refers to any hypothesis of these references. Please consider removal of the references from the abstract or, if required, use the complete reference and not just numbers.

Response: We have removed the references from the abstract.

2. Mechanical properties: The authors observe strengthening with a reduction of grain size and relate it to Hall-Petch hardening. According to the Petch-Stroh criterion this should be accompanied by an increase in ductility. I do not see any critical assessment of this apparent contradictory result. Simply stating that it is matter of ongoing investigations, does not seem sufficient for the reviewer.

Response: As the reviewer points out, the decrease in grain size can increase ductility, as smaller grains increase the number of grain boundaries that impede the movement of dislocations and promote plastic deformation. We have conducted a deeper analysis of the fracture surface, which reveals a dimpled structure for all three compositions, pointing to a ductile failure. However, in Mn50, there is a small fraction of the fracture surface that appears to be of brittle nature. Panels b-d has been added to Figure 4 as well as the following sentences to explain the reasoning:

Tensile testing was conducted to assess the mechanical performance of the as-printed MPEAs in a direction perpendicular to the build direction (Fig. 4). Notably, the yield strengths exhibited an upward trend, rising from 372.7 ± 10.8 MPa, 378.2 ± 11.6 MPa, to 411.9 ± 18.3 MPa in Mn40, Mn45, and Mn50, respectively. This enhancement in strength, particularly in Mn50, can be attributed to the grain refinement according to the Hall-Petch relationship. However, there is a slight decrease in ductility with increasing Mn content ($28.9 \pm 4.1\%$, $27.9 \pm 3.1\%$, and $26.8 \pm 1.0\%$, respectively). While the Petch-Stroh criterion suggests that the strengthening should be accompanied by increased ductility, the findings show otherwise. SEM analysis of the fracture surfaces reveals dimpled structures across all three compositions, indicating a ductile fracture (Fig. 4b-d). Though the fracture surface of Mn50 predominantly exhibits dimples, a very small portion shows a serrated fracture characteristic, which suggests a brittle failure in likely the sigma phase (Fig. 4d). While the sigma phase appears sporadically and does not seem to cause catastrophic embrittlement, observations suggest a still slight compromise in the overall ductility of the printed structure.

Fig. 4. Mechanical behavior and response of MPEAs. (a) Engineering stress vs strain curve of as-printed FeMnCoCr MPEAs. (b-d) SEM images of fracture surfaces of Mn40, Mn45, and Mn50, respectively.

Reviewer #2 (Remarks to the Author):

This work has assessed the microstructural and mechanical effect by consistently swapping Mn for Fe in a typical HEA system (FeMnCoCr) for DED. The striking finding was a significant refinement of the grain size and also an increase in strength with a trade-off in ductility when Mn was increased to 50%. The manuscript requires some clarification and strengthening before publication.

1. The bcc to fcc transformation is interesting, but the only physical evidence presented was a very brief capture of the bcc (211) reflection spot in the synchrotron data. The authors should show the volume fraction vs temperature plot for bcc and fcc. Hereby to strengthen the proposed mechanism. In addition, I was expecting to see (310) reflection as well. Is there any reason that it was not shown?

Response: Ideally, as the reviewer points out, we would quantify the bcc and fcc fraction from the synchrotron XRD data. However, due to the limitation of the setup, this analysis on the current dataset would not yield phase fractions accurate to the actual physical phenomenon. The limitations are the following: (1) the detectors only capture a portion of the diffracted cone, which only gives information of grains that happened to satisfy the Bragg's condition within a fairly narrow window of azimuths that the detectors cover, (2) the X-ray beam only covers a single projection that limits the number of grains that satisfy the Bragg's condition (as opposed to sample rotation, which would provide information about all grains). Given these limitations, we are unable to quantify phase fraction. However, we emphasize that multiple reflections of the bcc phase are detected in the GE detector shown in Extended Data Fig. 2, which captures a larger portion of the

diffracted cone at the expense of lower frequency. The Fe-bcc does not have a strong (310) reflection within the azimuths studied, which is the reason why we do not observe it in our dataset.

2. One further trend that I was confused was the reflection intensities for 311 and 222 in Mn40 and 45 vs 50 (Fig 1 b). They are not following the same trend in 2θ as 220 as it cools down from the melt, why? Rather, they show a linear transition from powder to cooling. Meanwhile, Mn50 was consistent with all the reflections shown.

Response: There are two factors about the 2θ vs time plots in Fig 1b that may have caused this confusion. Firstly, the linear transition from powder to cooling comes from diffraction from the powder surrounding the bead. In the setup, we used a powder bed to scan the laser to create a single track. Although a portion of the powder bed becomes melted and solidified into a bead, there remains residual powder on either side of the track, which has a uniform diffraction pattern (indicative of powder diffraction), although this information is lost when integrated along the azimuth as in Fig 1b. We are confident that this pattern comes from powder because the uniform powder diffraction pattern is clearly visible in the GE detector images, and evidence of this can be seen in the middle panels of Extended Data Fig. 2. Diffraction pattern that comes from the melted and solidifying bead is much spottier because individual grains are diffracting at certain spots along the azimuth. For example, if you took a look at the (220) ring in Mn45 in the middle panel in Extended Data Fig 2, there are two diffraction spots just to the left of the diffraction ring; the distinct spots come from grains in the melt (there is a shift in 2θ because it has been melted/heated), and the diffraction ring remains uniform from powder. The surrounding powder is heated up slightly from the laser source, which is why there is a slight shift in the 2θ values when integrated. Therefore, the continuous diffraction pattern in Mn40 and Mn45 comes from the surrounding powder. Such powder diffraction is less prominent in Mn50 likely because the powder bed used in Mn50 was slightly thinner, which allowed for a higher fraction of powder to be melted and solidified. The second reason for the apparent lack of diffraction data in Mn40 and Mn45 in the (311) and (222) reflections is because of their resulting grain sizes. In both compositions, the grains in the beads are fairly large, which limits the number of grains that can diffract onto the Debye Scherrer rings. On the other hand, Mn50 has smaller grains with more randomly oriented grains, which increases the chance of diffraction spots landing within the position of the detectors.

A faint diffraction pattern can be seen in the (311) reflection in both Mn40 and Mn45, which points to a relatively weaker signal from melted and solidified bead in these compositions as well. There is a branching of the reflections due to signal coming from the melted bead and surrounding powder, but the signal intensities from the bead are lower in Mn40 and Mn45 than those from the powder, which appears as a linear transition from powder to cooling. This is pointed out in the manuscript with the following sentences:

“The original powder peaks in Mn40 and Mn45 remain in the captured data even during melting (between $t \approx 3$ s and $t \approx 4.5$ s) due to diffraction from the residual powder surrounding the

melt pool and deposited bead. These powder peaks during melting are ignored for analysis and interpretation.”

3. Since oxides were frequently observed, the authors should disclose the oxygen level in the starting powder. Other relevant trace elements for C, S, N will be helpful, too. How was the chemical composition measured in the extended data? It needs to be disclosed.

Response: We ran ICP on the powders, whose results have now been added to the manuscript. Results show 0.0574 wt%, 0.0533 wt%, and 0.0364 wt% of oxygen in Mn40, Mn45, and Mn50 powders, respectively. In addition, the chemical composition in the extended data was measured via EDS, which has also been added to the title of the Extended Data Table 1.

4. I had a quick search in the literature, which I found other groups publishing on the same composition but did not see the same grain refinement trend by increasing Mn (with the same processing conditions, etc). What would be the reason? It deserves a good discussion. An example is here: <https://doi.org/10.3390/mi15010123>

Response: The paper provided by the reviewer considers $\text{Fe}_{30}\text{Mn}_{50}\text{Co}_{10}\text{Cr}_{10}$ and $\text{Fe}_{50}\text{Mn}_{30}\text{Co}_{10}\text{Cr}_{10}$ produced via powder bed fusion (PBF). However, as the reviewer points out, grain refinement is not reported between these materials. This is in contrast with our findings, which can be influenced due to the much higher solidification rates in PBF compared to DED, which was used in our study. Metastable phase formation and phase transformation are heavily influenced by cooling rates, and with cooling rates orders of magnitude higher in PBF may have resulted in a direct solidification into the stable fcc phase in both compositions fabricated via PBF. We also note that $\text{Fe}_{50}\text{Mn}_{30}\text{Co}_{10}\text{Cr}_{10}$ is a composition that was not used in our study; we, therefore, cannot rule out the possibility that this additional composition may have a different grain refinement mechanism that induces grain refinement on the same scale as metastable phase formation that we observed. While there are a few other studies on FeMnCoCr (<https://doi.org/10.1016/j.msea.2021.141264>), and some with additional minor/interstitial elements such as C (<https://doi.org/10.1016/j.addma.2023.103914>) and N/Ti (<https://doi.org/10.1080/21663831.2024.2340637>), these studies focus on a single composition containing low amounts of Mn to understand the deformation-induced fcc-to-hcp transformation. We note that, to our knowledge, our study is the first systematic investigation of the phase metastability during solidification in FeMnCoCr in AM. We added the following note at the end of the manuscript:

It is worth mentioning that the grain refinement has not been observed between MPEAs of similar composition, namely $\text{Fe}_{30}\text{Mn}_{50}\text{Co}_{10}\text{Cr}_{10}$ and $\text{Fe}_{50}\text{Mn}_{30}\text{Co}_{10}\text{Cr}_{10}$ fabricated via PBF⁴⁹. This finding suggests that the grain refinement mechanism is highly dependent on the solidification rate, which is higher in PBF than in DED. The effect of solidification rates on metastable phase formation is subject to further study.

5. The authors should also provide sufficient level of characterisation for powder microstructure, such as the starting grain size, to make the story more convincing.

Response: At the recommendation of the reviewer, we have acquired additional characterization of powder microstructure. The following BSE and SE images have been added to the manuscript as Extended Data Figure 3, along with the following text:

Comparable grain sizes in their powder form (Extended Data Fig. 3) suggest that differences in solidification pathways cause the grain refinement. The powders are all fully fcc based on the synchrotron XRD results.

Extended Data Fig. 3. Backscatter electron (BSE) and secondary electron (SE) images of the cross-sections of representative powder particles of Mn40, Mn45, and Mn50. These powders are fully fcc based on synchrotron XRD results.

6. I don't think as-cast structure in Fig 4 should be there, as the comparison is not very fair between

casting vs DED. As the literature in AM has shown higher yield strength of the same composition, such as the one above (LPBF though).

Response: The mechanical behavior of the as-cast MPEA was initially added to highlight the impact of AM processing on the mechanical properties; however, it has now been removed to focus on the mechanical properties of the three as-printed compositions.

RESPONSE TO REVIEWERS' COMMENTS

Akane Wakai, Jenniffer Bustillos, Noah Sargent, Jamesa Stokes, Wei Xiong, Tim Smith, and Atieh Moridi

Dear Reviewers,

Once again, we greatly appreciate both reviewer's thoughtful comments. Following reviewer 1's recommendations, we made a modification to the manuscript to clarify that the reasons for the reduced ductility are currently unknown. Please find our response below and changes to the manuscript highlighted in yellow.

Respectfully,

Atieh Moridi

Reviewer #1 (Remarks to the Author):

The authors have provided additional information on the fracture behavior but do not provide a compelling theory on the observed reduction in ductility. While I do not see, how this can be argued simply ignoring the absence of an attempt remains insufficient. If the authors do not want to claim a responsible effect, they should state that there is presently no explanation for the observed behavior.

Response: We understand the reviewer's perspective on the level of detail on the fracture behavior in the manuscript. While the deformation mechanisms are interesting, we have decided to place the focus of this study on the solidification behavior and microstructure. We have thus added the sentence "There is currently no explanation of the observed trend." following the reviewer's recommendation.

Reviewer #2 (Remarks to the Author):

The authors have addressed my questions appropriately and revised the manuscript accordingly. The overall methodology was elegant and I am convinced the conclusions are sound. I would encourage the authors to disclose when the delta phase disappeared from the diffraction spots on the GE detector. Currently it was only termed as 'quickly disappear' where quantifying the temporal evolution, despite a much lower acquisition (and uncertainty), would help estimate the transformation temperature and interpret the nature of the transformation -- whether it was fully solid state or involving liquid as proposed.

By way of readability, some optional suggestions here. Consider moving Fig 1a to extended data -- this was not so critical in my opinion and it has caused some confusion as I read. You can also consider moving extended data Mn50 evolution to Fig 1 as that is more convincing with rings of delta reflections.

Response: We greatly appreciate the reviewer's support of the manuscript. While we would ideally be able to capture a larger portion of the diffracted cone at a faster rate on the GE detector, current physical limitations (a low signal-to-noise ratio and sample-to-detector distance that limits the degree of peak separation enough for peak identification) prevent us from acquiring a more statistically sound dataset on large area detectors at a higher frequency. Unfortunately, at the acquisition rate of 4Hz, the temporal resolution does not enable us to get an accurate estimate of the time of phase appearance and transformation. In addition, temperature is very difficult to estimate from operando X-ray diffraction due to contributing factors such as residual stress and chemical gradients that also alter the diffraction behavior. Due to these considerations, we have decided not to quantify the time frames from the dataset on the GE detector.

As for Fig 1a and Extended Data Fig. 2, we feel that Fig 1a presents a critical portion of the alloy design process, which is the thermodynamic simulation. Although it has its limitations, the metastable phase formation predicted via simulations gives us an important trend, which is why we have incorporated it into the main text. Although the GE detector images on Extended Data Figure 2 are also important, it gives further proof at a much lower time resolution than Eiger detector images, which is why we have decided to incorporate them as Extended Data.

RESPONSE TO REVIEWERS' COMMENTS

Akane Wakai, Jenniffer Bustillos, Noah Sargent, Jamesa Stokes, Wei Xiong, Tim Smith, and Atieh Moridi

Dear Reviewers,

We greatly appreciate both reviewer's thoughtful comments. We hope to clarify some of the confusions and questions and explain why we believe our present work would be of interest to the readers of Nature Communications after this revision. In addition, some word choices have been replaced following suggestions from the reviewers. We strongly believe that the new word choices convincingly propose solidification pathway engineering as a new parameter for microstructural engineering through the combination of our in-situ synchrotron studies and microstructural evaluation. We are writing to ask if you would be willing to kindly give us an opportunity to make such revisions for publication at Nature Communications. Please find our response below and changes to the manuscript highlighted in yellow.

Very Respectfully,

Atieh Moridi

Reviewer #1 (Remarks to the Author):

The present manuscript reports on alloying-related effects in a multi-principal element system and its interactions with additive manufacturing conditions. The reported results are new and interesting and the conclusions drawn are presented in a compelling manner. There remain some minor thoughts outlined below that the authors should consider prior to acceptance of the article.

1. It is unusual to use references in the abstract unless the work specifically refers to any hypothesis of these references. Please consider removal of the references from the abstract or, if required, use the complete reference and not just numbers.

Response: The original submission was to Nature, which asked for a fully referenced summary- hence it has references. We have removed the references from the abstract.

2. Mechanical properties: The authors observe strengthening with a reduction of grain size and relate it to Hall-Petch hardening. According to the Petch-Stroh criterion this should be accompanied by an increase in ductility. I do not see any critical assessment of this apparent contradictory result. Simply stating that it is matter of ongoing investigations, does not seem sufficient for the reviewer.

Response: As the reviewer points out, the decrease in grain size can increase ductility, as smaller grains increases the number of grain boundaries that impede the movement of dislocations and promote plastic deformation. We have conducted a deeper analysis into the fracture surface, which reveals a dimpled structure for all three compositions, pointing to a ductile failure. However, in Mn50, there is a small fraction of the fracture surface that appears to be of brittle nature. The following panels has been added to Figure 4 as well as the following sentences:

Tensile testing was conducted to assess the mechanical performance of the as-printed MPEAs in a direction perpendicular to the build direction (Fig. 4). Notably, the yield strengths exhibited an upward trend, rising from 372.7 ± 10.8 MPa, 378.2 ± 11.6 MPa, to 411.9 ± 18.3 MPa in Mn40, Mn45, and Mn50, respectively. This enhancement in strength, particularly in Mn50, can be attributed to the grain refinement according to the Hall-Petch relationship. However there is a slight decrease in ductility with increasing Mn content ($28.9 \pm 4.1\%$, $27.9 \pm 3.1\%$, and $26.8 \pm 1.0\%$, respectively). While the Petch-Stroh criterion suggests that the strengthening should be accompanied by increased ductility, the findings show otherwise. SEM analysis of the fracture surfaces reveals dimpled structures across all three compositions, indicating a ductile fracture (Fig. 4b-d). Though the fracture surface of Mn50 predominantly exhibits dimples, a very small portion shows a serrated fracture characteristic, which suggests brittle failure in likely the sigma phase (Fig. 4d). While the sigma phase appears sporadically and does not seem to cause catastrophic embrittlement, observations suggest a still slight compromise in the overall ductility of the printed structure.

Fig. 4. Mechanical behavior and response of MPEAs. (a) Engineering stress vs strain curve of as-printed FeMnCoCr MPEAs. (b-d) SEM images of fracture surfaces of Mn40, Mn45, and Mn50, respectively.

Reviewer #2 (Remarks to the Author):

This work has assessed the microstructural and mechanical effect by consistently swapping Mn for Fe in a typical HEA system (FeMnCoCr) for DED. The striking finding was a significant refinement of the grain size and also an increase in strength with a trade-off in ductility when Mn was increased to 50%. The manuscript requires some clarification and strengthening before publication.

1. The bcc to fcc transformation is interesting, but the only physical evidence presented was a very brief capture of the bcc (211) reflection spot in the synchrotron data. The authors should show the volume fraction vs temperature plot for bcc and fcc. Hereby to strengthen the proposed mechanism. In addition, I was expecting to see (310) reflection as well. Is there any reason that it was not shown?

Response: Ideally, as the reviewer points out, we would quantify the bcc and fcc fraction from the synchrotron XRD data. However, due to the limitation of the setup, this analysis on the current dataset would not yield phase fractions accurate to the actual physical phenomenon. The limitations are the following: (1) the detectors only capture a portion of the diffracted cone, which only gives information of grains that happened to satisfy the Bragg's condition within a fairly narrow window of azimuths that the detectors cover, (2) the X-ray beam only covers a single projection that limits the number of grains that satisfy the Bragg's condition (as opposed to sample rotation, which would provide information about all grains). Given these limitations, we are unable to quantify phase fraction. However, we emphasize that multiple reflections of the bcc phase are detected in the GE detector shown in Extended Data Fig. 2, which captures a larger portion of the diffracted cone at the expense of lower frequency. The Fe-bcc does not have a strong (310) reflection within the azimuths studied, which is the reason why we do not observe it in our dataset.

2. One further trend that I was confused was the reflection intensities for 311 and 222 in Mn40 and 45 vs 50 (Fig 1 b). They are not following the same trend in 2θ as 220 as it cools down from the melt, why? Rather, they show a linear transition from powder to cooling. Meanwhile, Mn50 was consistent with all the reflections shown.

Response: There are two factors about the 2θ vs time plots in Fig 1b that may have caused this confusion. Firstly, the linear transition from powder to cooling comes from diffraction from the powder surrounding the bead. In the setup, we used a powder bed to scan the laser to create a single track. Although a portion of the powder bed becomes melted and solidified into a bead, there remains residual powder on either side of the track, which has a uniform diffraction pattern (indicative of powder diffraction), although this information is lost when integrated along the azimuth as in Fig 1b. We are confident that this pattern comes from powder because the uniform powder diffraction pattern is clearly visible in the GE detector images, and evidence of this can be seen in the middle panels of Extended Data Fig. 2. Diffraction pattern that comes from the melted and solidifying bead is much spottier because individual grains are diffracting at certain spots along the azimuth. For example, if you took a look at the (220) ring in Mn45 in the middle panel, there are two diffraction spots just to the left of the diffraction ring; the distinct spots come from grains in the melt (there is a shift in 2θ because it has been melted/heated), and the diffraction ring remains uniform from powder. The surrounding powder is heated up slightly from the laser source, which is why there is a slight shift in the 2θ values when integrated. Therefore, the continuous diffraction pattern in Mn40 and Mn45 comes from the surrounding powder. Such powder diffraction is less prominent in Mn50 likely because the powder bed used in Mn50 was slightly thinner, which allowed for a higher fraction of powder to be melted and solidified. The second reason for the apparent lack of diffraction data in Mn40 and Mn45 in the (311) and (222) reflections is because of their resulting grain sizes. In both compositions, the grains in the beads are fairly large, which limits the number of grains that can diffract onto the Debye Scherrer rings. On the other hand, Mn50 has smaller grains with more randomly oriented grains, which increases the chance of diffraction spots landing within the position of the detectors.

3. Since oxides were frequently observed, the authors should disclose the oxygen level in the starting powder. Other relevant trace elements for C, S, N will be helpful, too. How was the chemical composition measured in the extended data? It needs to be disclosed.

Response: We ran ICP on the powders, whose results have now been added to the manuscript. Results show 0.0574 wt%, 0.0533 wt%, and 0.0364 wt% of oxygen in Mn40, Mn45, and Mn50 powders, respectively. In addition, the chemical composition in the extended data was measured via EDS, which has also been added to the title to the Extended Data Table 1.

4. I had a quick search in the literature, which I found other groups publishing on the same

composition but did not see the same grain refinement trend by increasing Mn (with the same processing conditions, etc). What would be the reason? It deserves a good discussion. An example is here: <https://doi.org/10.3390/mi15010123>

Response: The paper provided by the reviewer considers $\text{Fe}_{30}\text{Mn}_{50}\text{Co}_{10}\text{Cr}_{10}$ and $\text{Fe}_{50}\text{Mn}_{30}\text{Co}_{10}\text{Cr}_{10}$ produced via powder bed fusion (PBF) and touches on important mechanical behaviors. However, as the reviewer points out, grain refinement is not reported between these materials. This is in contrast with our findings, which can be influenced due to the much higher solidification rates in PBF compared to DED, which was used in our study. Metastable phase formation and phase transformation is heavily influenced by cooling rates, and with cooling rates orders of magnitude higher in PBF may have resulted in a direct solidification into the stable fcc phase in both compositions fabricated via PBF. We also note that $\text{Fe}_{50}\text{Mn}_{30}\text{Co}_{10}\text{Cr}_{10}$ is a composition that was not used in our study; we therefore cannot rule out the possibility that this additional composition may have a different grain refinement mechanism that induces grain refinement on the same scale as metastable phase formation that we observed. We added the following note at the end of the manuscript:

It is worth mentioning that the grain refinement has not been observed between MPEAs of similar composition, namely $\text{Fe}_{30}\text{Mn}_{50}\text{Co}_{10}\text{Cr}_{10}$ and $\text{Fe}_{50}\text{Mn}_{30}\text{Co}_{10}\text{Cr}_{10}$ fabricated via PBF⁴⁹. This finding suggests that the grain refinement mechanism is highly dependent on the solidification rate, which is higher in PBF than in DED. The effect of solidification rates on metastable phase formation is subject to further study.

5. The authors should also provide sufficient level of characterisation for powder microstructure, such as the starting grain size, to make the story more convincing.

Response: At the recommendation of the reviewer, we have acquired additional characterization of powder microstructure. The following BSE and SE images have been added to the manuscript as Extended Data Figure 3, along with the following text:

Comparable grain sizes in their powder form (Extended Data Fig. 3) suggests that the cooling rates in DED causes the grain refinement during solidification and transformation.

Extended Data Fig. 3. Backscatter electron (BSE) and secondary electron (SE) images of the cross-sections of representative powder particles of Mn40, Mn45, and Mn50.

6. I don't think as-cast structure in Fig 4 should be there, as the comparison is not very fair between casting vs DED. As the literature in AM has shown higher yield strength of the same composition, such as the one above (LPBF though).

Response: The mechanical behavior of the as-cast MPEA was initially added to highlight the impact of AM processing on the mechanical properties; however, it has now been removed to focus on the mechanical properties of the three as-printed compositions. In addition, the slight embrittlement in Mn50 has been highlighted as recommended by Reviewer 1.

Dear Editor and Reviewers,

Once again, we greatly appreciate both reviewer's thoughtful comments. Following reviewer 1's recommendations, we made a modification to the manuscript to clarify that the reasons for the reduced ductility are currently unknown. Please find our response below and changes to the manuscript highlighted in yellow.

Respectfully,

Atieh Moridi

Reviewer #1 (Remarks to the Author):

The authors have provided additional information on the fracture behavior but do not provide a compelling theory on the observed reduction in ductility. While I do not see, how this can be argued simply ignoring the absence of an attempt remains insufficient. If the authors do not want to claim a responsible effect, they should state that there is presently no explanation for the observed behavior.

Response: We understand the reviewer's perspective on the level of detail on the fracture behavior in the manuscript. While the deformation mechanisms are interesting, we have decided to place the focus of this study on the solidification behavior and microstructure. We have thus added the sentence "There is currently no explanation of the observed trend." following the reviewer's recommendation.

Reviewer #2 (Remarks to the Author):

The authors have addressed my questions appropriately and revised the manuscript accordingly. The overall methodology was elegant and I am convinced the conclusions are sound. I would encourage the authors to disclose when the delta phase disappeared from the diffraction spots on the GE detector. Currently it was only termed as 'quickly disappear' where quantifying the temporal evolution, despite a much lower acquisition (and uncertainty), would help estimate the transformation temperature and interpret the nature of the transformation -- whether it was fully solid state or involving liquid as proposed.

By way of readability, some optional suggestions here. Consider moving Fig 1a to extended data -- this was not so critical in my opinion and it has caused some confusion as I read. You can also consider moving extended data Mn50 evolution to Fig 1 as that is more convincing with rings of delta reflections.

Response: We greatly appreciate the reviewer's support of the manuscript. While we would ideally be able to capture a larger portion of the diffracted cone at a faster rate on the GE detector, current physical limitations (a low signal-to-noise ratio and sample-to-detector distance that limits the degree of peak separation enough for peak identification) prevent us from acquiring a more statistically sound dataset on large area detectors at a higher frequency. Unfortunately, at the acquisition rate of 4Hz, the temporal resolution does not enable us to get an accurate estimate of the time of phase appearance and transformation. In addition, temperature is very difficult to estimate from operando X-ray diffraction due to contributing factors such as residual stress and chemical gradients that also alter the diffraction behavior. Due to these considerations, we have decided not to quantify the time frames from the dataset on the GE detector.

As for Fig 1a and Extended Data Fig. 2, we feel that Fig 1a presents a critical portion of the alloy design process, which is the thermodynamic simulation. Although it has its limitations, the metastable phase formation predicted via simulations gives us an important trend, which is why we have incorporated it into the main text. Although the GE detector images on Extended Data Figure 2 are also important, it gives a redundant information at a much lower time resolution than Eiger detector images, which is why we have decided to incorporate them as Extended Data.